# Immunogenicity of Novel AAV Capsids for Retinal Gene Therapy

**DOI:** 10.3390/cells11121881

**Published:** 2022-06-09

**Authors:** Miranda Gehrke, Maria Diedrichs-Möhring, Jacqueline Bogedein, Hildegard Büning, Stylianos Michalakis, Gerhild Wildner

**Affiliations:** 1Department of Ophthalmology, University Hospital, LMU Munich, Mathildenstr. 8, 80336 Munich, Germany; miranda.gehrke@med.uni-muenchen.de (M.G.); maria.diedrichs-moehring@med.uni-muenchen.de (M.D.-M.); jacqueline.bogedein@med.uni-muenchen.de (J.B.); 2Institute of Experimental Hematology, Hannover Medical School, Carl-Neuberg-Straße 1, 30625 Hannover, Germany

**Keywords:** antibodies, neutralization, cellular immunity, innate immunity, adaptive immunity, monocyte-derived DC, immunofluorescence, cytokines, chemokines

## Abstract

Objectives: AAV vectors are widely used in gene therapy, but the prevalence of neutralizing antibodies raised against AAV serotypes in the course of a natural infection, as well as innate and adaptive immune responses induced upon vector administration, is still considered an important limitation. In ocular gene therapy, vectors applied subretinally bear the risk of retinal detachment or vascular leakage. Therefore, new AAV vectors that are suitable for intravitreal administration for photoreceptor transduction were developed. Methods: Here, we compared human immune responses from donors with suspected previous AAV2 infections to the new vectors AAV2.GL and AAV2.NN—two capsid peptide display variants with an enhanced tropism for photoreceptors—with the parental serotype AAV2 (AAV2 WT). We investigated total and neutralizing antibodies, adaptive and innate cellular immunogenicity determined by immunofluorescence staining and flow cytometry, and cytokine secretion analyzed with multiplex beads. Results: While we did not observe obvious differences in overall antibody binding, variants—particularly AAV2.GL—were less sensitive to neutralizing antibodies than the AAV2 WT. The novel variants did not differ from AAV2 WT in cellular immune responses and cytokine production in vitro. Conclusion: Due to their enhanced retinal tropism, which allows for dose reduction, the new vector variants are likely to be less immunogenic for gene therapy than the parental AAV2 vector.

## 1. Introduction

Most of the current in vivo gene therapy approaches rely on adeno-associated virus (AAV) vectors as a delivery tool [1]. Although high vector doses must be applied, safety and efficacy data fostered three market approvals of AAV-vector-based gene therapies [2], and more are expected in the near future; they will mostly target diseases of the central nervous system, muscle, liver, and the eye [3].

AAV vectors are composed of an icosahedral capsid containing a single-stranded (ss) DNA genome. For efficient transduction, AAV vectors must bind to receptors on their target cells, followed by internalization mediated by specific capsid residues and differing between serotypes [4,5]. The capsid of the best-studied natural serotype, AAV2, interacts with heparan sulfate proteoglycan (HSPG) and co-receptors such as avβ5 or α5β1 integrins [4,6]. In addition, AAVR and GPR108, which interact with distinct parts of the capsids, are host factors that are essential for efficient cell transduction of nearly all AAV serotypes [7]. Capsid engineering makes it possible to alter the host–vector interaction and, thus, to modify cell tropism, enhance intracellular transport, or decrease host immune activation [8,9,10].

While AAV vectors generally have less immunogenic potential than other viral vectors (e.g., adenovirus), transient and dose-dependent immune responses are commonly observed in preclinical and clinical studies [11,12,13]. High intravenously applied doses, for example, can induce immune responses and have been discussed as a cause for fatal complications in a recent human clinical trial on XLMTM [14]. 

Therefore, whenever possible, local application of AAV vectors is preferrable, such as in ocular gene therapy, where the vectors at relatively low doses are either injected intravitreally or subretinally, depending on the location of the retinal target cells and the AAV serotype used [15]. Nevertheless, also following intravitreal injection, intraocular inflammation (uveitis) of both treated and even untreated eyes has been observed [16,17,18]. In contrast, injection of AAV vectors into the subretinal space is less burdened with intraocular inflammation but increases the risk of uncontrolled retinal detachment or leakage of retinal vessels [19,20]. 

Immune reactions to viral vectors can be triggered by the capsids, the genome, and the transgene products, while other factors, such as the route of administration and the vector dose, influence the extent of inflammation [6,21]. Thus, there is a need for vectors with improved transduction efficiency and target cell selectivity to reduce vector dose and, thus, the antigenic load, which would ideally be combined with modifications that lower vector immunogenicity. Optimized vectors should evade the host’s immune response while possessing enhanced tissue tropism and transduction efficiency [8,22].

Here, we investigated the antibody responses of donors without prior gene therapy for preexisting total and neutralizing antibody responses to the wildtype AAV2 (AAV2 WT) and the variants AAV2.GL and AAV2.NN. Moreover, we looked at cellular immune responses in vitro and investigated the proliferation and cytokine/chemokine secretion of immune cell populations in response to AAV2 WT compared to the capsid variants. While the total antibody responses did not differ between the capsids, the novel variants were less efficiently neutralized than AAV2 WT. We could not detect specific adaptive cellular immune responses or cytokine or chemokine secretion from peripheral blood cells to AAV2 WT or the variants, but we found an increase in IFN-β production by CD14+ monocytes and CD11c+ DCs, as well as an increase in CD11c+/CD14+ monocyte-derived DCs producing IL-1β and IFN-β in response to LPS and the AAV2 capsids, with no difference between WT and the variants. Although no overall reduced immunogenicity could be demonstrated for the new variants, their enhanced cellular tropism allows one to use lower vector doses for gene therapy compared to the WT, which has the potential to reduce the risk of immune stimulation.

## 2. Materials and Methods

### 2.1. Human Blood Donors

The collection of peripheral blood for the analysis of antibody or cellular immune responses from 24 anonymized human donors with informed consent was approved by the local ethics committee (Project number 227/03). None of the donors (*n* = 24) had previously received a gene therapy with AAV vectors. The median age was 41 years, with a range of 50 years (oldest: 61 years to youngest: 11 years). Plasma and peripheral blood mononuclear cells (PBMCs) were obtained from heparinized blood and serum from coagulated blood samples. PBMCs were separated by a Ficoll-Hypaque 1077 gradient (Merck, Darmstadt, Germany) and stimulated in vitro with the indicated AAV vectors, as well as with recall antigens: TT (tetanus toxoid), PPD (tuberculin; both gifts from Aventis, Marburg, Germany), or LPS (lipopolysaccharide, Invivogen, San Diego, CA, USA). Seven of the 24 donors (Table 1) were analyzed for their cellular reactivity. In vitro expansion of cell populations was determined with immunofluorescence staining followed by FACS analysis; cytokine secretion of culture supernatants was investigated in a bioplex bead assay (see below).

### 2.2. AAV Vectors

Recombinant adeno-associated virus (AAV) vectors carrying a self-complementary AAV genome with a gene expression cassette coding for the enhanced green fluorescent protein (eGFP) under control of the human cytomegalovirus promoter (CMV) and followed by a Simian vacuolating virus 40 (SV40) polyadenylation signal were produced with AAV2 wildtype (AAV2 WT), AAV2.GL, or AAV2.NN capsids [23] according to previously described procedures [24]. AAV vector preparations were stored at −80 °C until use. 

### 2.3. Determination of Capsid Titers of AAV2 Preparations

Capsid titers were determined via an AAV2 Titration ELISA Kit 2.0R (Progen, Heidelberg, Germany). The capsids were used in respective concentrations as bait for the ELISA to determine total binding antibodies in human sera and for the PBMC stimulation experiments. The capsid titers were also needed to calculate the neutralization capacity of antibodies in relation to the transduction rate of the AAV vectors. Inter/intraassay variability for ELISA was 12.4% and 11.23%, respectively.

### 2.4. Antibody Responses to AAV2 Capsids

To determine antibody responses to the various AAV vectors, an ELISA was established by directly coating the AAV capsids in coating buffer (Candor Bioscience GmbH, Wangen, Germany) overnight at 4 °C to ELISA plates (MaxiSorp; Nunc Nalgene/Merck, Heidelberg, Germany) followed by 3 h of incubation at RT with blocking solution (Candor). The mouse monoclonal antibody A20 (stock 50 µg/mL, Progen, Heidelberg, Germany) that bound to AAV2 as well as to AAV2.GL and AAV2.NN was used to determine the optimal capsid concentration for coating the ELISA plates. The binding of A20 to the capsid coat was determined with a biotinylated anti-mouse IgG (ThermoFisher Scientific, Erlangen, Germany) and developed with HRP-conjugated streptavidin and Turbo TRMB-ELISA Substrate Solution (both ThermoFisher Scientific, Erlangen, Germany). The optimal concentration of 6.9 × 10^7^ per well for each capsid was used as a coat for further testing of the total antibody binding with human sera. The sera were added in duplicate dilutions from 1:20 to 1:2000 in LowCross buffer (Candor Bioscience GmbH, Wangen, Germany) to the capsid coat and incubated for 1 h at RT. As a secondary antibody, a biotinylated goat anti-human Ig (IgG/IgM/IgA) antibody (1:20,000; ThermoFisher Scientific, Erlangen, Germany) was used and developed with HRP-conjugated streptavidin (1:2000) and substrate solution as described above. 

The mouse monoclonal antibody A20 was used as a positive control. The OD 450 nm of the negative control (no primary antibody) was subtracted from the samples’ ODs. ODs were plotted against the serum dilutions, and the OD 450 for each serum and AAV2 capsid was determined for the 1:400 dilution.

### 2.5. Neutralizing Antibody Assay

HeLa cell cultures were grown in Dulbecco’s modified Eagle’s medium (DMEM), low glucose with GlutaMAX™ (Merck, Darmstadt, Germany), supplemented with 50 U penicillin/50 mg streptomycin, 1 mM sodium pyruvate, and 10% fetal calf serum (FCS) (all from Merck, Darmstadt, Germany) in 95% humidified atmosphere containing 5% CO_2_ at 37 °C.

To test the neutralizing capacity of the human sera, HeLa cells were distributed in 12-well plates (Nunc) at a density of 50,000 cells/well in 1 mL of DMEM. The cells were cultured for 24 h at 37 °C and 5% CO_2_ before the serum dilutions, pre-incubated for 1 h at RT with previously defined concentrations for each AAV preparation, were added and incubated with the HeLa cells. The serum dilutions used for the neutralization assay were 1:20, 1:200, 1:1000, 1:2000, 1:5000, 1:25,000, and 1:125,000. The MOI (multiplicity of infection, defining the ratio of vector particles to their target cells, determining the number of vector particles to obtain similar target cell transduction for different vector preparations) was determined for each AAV2 vector prior to the neutralization assay. The MOI for AAV2 WT was 20, that for AAV2.GL was 200, and that for AAV2.NN was 250. The differences in the genome content of the vector preparations (AAV2 WT: 9.47%; AAV2.GL: 14.25%; AAV2.NN: 15.27%) were considered for the calculation of the serum dilution needed for 50% neutralization. The neutralization differences appearing from the different vector genome contents were adjusted as follows: reciprocal serum dilution/(percentage of capsids with genome × 100). Medium only or AAV dilutions without human serum were used as controls. After 48 h, HeLa cells were harvested by trypsinization and fixed in 4% PFA on ice. The number of eGFP-positive cells was determined using flow cytometry (Cyoflex S, Beckman Coulter, Krefeld, Germany; FACSCalibur, BD, Heidelberg, Germany) and analyzed with FlowJo 10.5.0 software BD, Berlin/Heidelberg, Germany). 

The neutralizing antibody titer was determined by calculating the highest dilution that inhibited AAV transduction by 50% (N50) compared to vectors without serum/plasma. Samples were considered as neutralizing when the serum titer (N50) was 1:40 or higher, since higher serum concentrations induced non-specific inhibition of vector transduction. The interassay variability for the neutralization assay was 17.4%. 

### 2.6. In Vitro Stimulation of PBMCs

PBMCs were cultured in DMEM (Merck, Darmstadt, Germany) with 4500 mg/L glucose, supplemented with 4 mM L-glutamine, 1 mM sodium pyruvate, 36 µg/mL L-asparagine, MEM non-essential amino acids, and 50 U penicillin/50 mg streptomycin (all from Merck, Darmstadt, Germany), as well as either 5% pooled human serum from at least 20 healthy, male donors below the age of 30 years or 5% Panexin CD as a serum supplement (Pan-Biotech, Aidenbach, Germany); they were cultivated at 37 °C in a humidified atmosphere with 7% CO_2_ as described below. 

### 2.7. Cytokine Bioplex Assay

PBMCs were cultured as triplicates of 2 × 10^5^–8 × 10^5^ cells/well in 96-well plates (TPP, Trasadingen, Switzerland) with AAV vector (2.74 × 10^8^ capsids/µL) and PPD (Purified Protein Derivative of M. tuberculosis, tuberculin), as well as tetanus toxoid (TT), respectively (both 10 µg/mL), as positive controls, and cells in medium without antigen as negative control. Supernatants were collected from PBMC cultures stimulated as described above after 24, 48, 72, and 96 h to include both earlier and later secreted cytokines and chemokines before they were used by the cells. The supernatants were pooled in equal volumes immediately prior of the cytokine secretion analysis with a Bioplex-Assay kit (Bio-RAD, Feldkirchen, Germany) that detected 48 cytokines and chemokines according to the manufacturer’s instructions and analyzed with a Bio-Plex Reader and the Bio-Plex Manager software (Bio-RAD, Feldkirchen, Germany). The kit included analytes for β-NGF, CTACK/CCL27, Eotaxin/CCL11, FGF-basic, G-CSF, GM-CSF, GRO-α (Gro-a/KC/CXCL1), HGF, IFN-α2, IFN-γ, IL-1α, IL-1Rα, IL-2Rα, IL-1β, IL-2, IL-3, IL-4, IL-5, IL-6, IL-7, IL-8/CXCL8, IL-9, IL-10, IL-12(p40), IL-12(p70), IL-13, IL-15, IL-16, IL-17A, IL-18, IP-10/CXCL10, LIF, M-CSF, MCP-1/CCL2, MCP-3/CCL7, MIG/CXCL9, MIP-1α/CCL3, MIP-1β/CCL4, MIF, PDGF-BB, RANTES/CCL5, SCF, SCGF-β, SDF-1α, TRAIL, TNF-α, TNF-β, and VEGF-A.

The amounts of cytokines (pg/mL) were calculated from the median fluorescence value of at least 50 beads measured per analyte and sample. 

### 2.8. AAV-Induced Expansion of Cell Populations and Intracellular Cytokine Expression

PBMCs were cultured at densities of 1.5–4 × 10^6^ per well in the presence of 6 × 10^9^ AAV capsids/well/mL for either 24 h or four days, as mentioned above. Medium only and LPS (Invivogen, Toulouse, France; 5 µg/mL) were used as controls. Surface and intracellular immunofluorescence staining of cells was performed after 24 h and 4 days as previously described [25]. Following fluorochrome-coupled antibodies were used according to the manufacturer’s instructions: CD3 (clone OKT3, APC), CD4 (clone RPA-T4, FITC), CD8 (clone RPA-T8, PE), CD69 (clone FN50, PerCP/Cy5), CD56 (clone 5.1H11, FITC), CD19 (clone 4G7, PE), CD14 (clone 63D3, PerCP/Cy5), and CD11c (clone 3.9, PE) (all from Biolegend, San Diego, CA, USA). Mouse serum (3%) was added to block unspecific Fc-receptor binding of the antibodies. For intracellular cytokine detection, cells were permeabilized and fixed for 45 min using the eBioscience permeabilization kit (ThermoFisher Scientific, Erlangen, Germany) according to the manufacturer’s instruction and stained with FITC-coupled anti-IFN-β (clone MMHB-3, Biozol, Eching, Germany) and APC-coupled anti-IL-1β (clone 8516, BioTechne, Minneapolis, MN, USA) monoclonal antibodies. Data were acquired with a FACScalibur (BD, Heidelberg, Germany) and analyzed with the FlowJo 10.5.0 software (BD, Heidelberg, Germany). The cut-off for the different fluorescence intensities was set according to the respective staining with isotype controls.

### 2.9. Statistics

The Friedman test and Dunn’s Multiple Comparison test were performed to calculate the significance for all assays, assuming that data were not normally distributed. *P* values are only provided for significant differences. 

## 3. Results

### 3.1. Similar Total Antibody Binding Abilities for AAV2 WT, AAV2.GL, and AAV2.NN

To characterize the presence and specificity of AAV-reactive antibodies, serum or plasma from 24 donors was assayed by indirect ELISA, using a reference serum (H2) with proven intermediate levels of anti-AAV2 antibodies or buffer as positive and negative controls, respectively (Figure 1). Donors were considered AAV2-seropositive if the antibody concentration (OD) was greater than the reference serum H2 at a dilution of 1:400. Among the tested population, a seroprevalence for AAV2 of 91.7% was observed (22 of the 24 assayed donors), and the positive sera showed equal binding affinity to each of the three tested capsids. No significant difference in antibody binding was observed between the capsid variants (Figure 1A), suggesting that the antibodies elicited during previous infections with naturally occurring AAV2 are cross-reactive between wildtype and variants. This indicates that the peptide insertions did not influence the general binding of naturally induced anti-AAV2 WT antibodies.

### 3.2. Decreased Neutralizing Antibody Binding to the AAV2.GL Mutant

The neutralizing capacity of the antibodies was tested through preincubation of eGFP-encoding vectors with human sera and subsequent transduction of HeLa cells, followed by quantification of the eGFP-expressing cell fraction with FACScan analysis. Data are presented as the dilution of samples at 50% neutralization of the vectors (IC_50_). A dilution of 1:40 or higher was considered seropositive. Variant AAV2.GL was significantly less neutralized than the wildtype (*p* < 0.001) and variant AAV2.NN (*p* < 0.01) (Figure 1B). Although the differences in the neutralization between WT and variant AAV2.NN did not reach statistical significance, a trend was clearly evident even for this variant. FACS analysis of eGFP + HeLa cells after transduction with AAV2 WT, AAV2.NN, and AAV2.GL vectors, which had been pre-incubated with a 1:200 or a 1:2000 dilution of serum U103, is shown in Figure 1C. At a 1:2000 dilution, both genetically modified capsids showed less neutralization than AAV2 WT. The eGFP+ fraction of HeLa cells was 23.5% after transduction with AAV2.GL, which was preincubated with serum U103; the corresponding values were 18.7% for preincubated AAV2.NN and only 9.7% for preincubated AAV2 WT (Figure 1C). The lower dilution (1:200) of serum U103 led to an almost complete neutralization of all three vectors, but nevertheless, the transduction of HeLa cells was 3- and 1.4-fold higher with AAV2.GL and AAV2.NN compared to AAV2 WT, indicating an escape of the modified capsids from neutralization (Figure 1C). 

A total of 65.2% of the serum samples (15 of 23, cutoff at a dilution of >1:40) had neutralizing antibodies against AAV2, and a mean serum dilution of 1:147 was needed to achieve half-maximal neutralization of AAV2 WT (Figure 1D). Lower dilutions were needed to achieve the same level of neutralization of AAV2.NN (1:102) and AAV2.GL (1:75). 

There was no correlation between the titers of total anti-capsid antibodies and their neutralization capacity (Figure 1D). Generally, there was a high interindividual variability with sera showing high total antibodies and little or even no neutralization capacity and vice versa (Figure 1D); e.g., donors H1 and H2, who were also used for cellular assays, had comparable titers of total antibodies, while only donor H2 also had neutralizing antibodies.

There was also no obvious difference in the age distribution between low, intermediate, and strong binders. Donors with the lowest antibody responses (H7, U1, U100, U51, U103, H4) were only slightly younger (mean 35.6 years) than those with an intermediate (U11, H6, U34, H3, U43, N2, H5, U45, H2, H1; mean: 43.9 years) or strong (U13, N3, N1, U49, N4, HP, U73; mean: 39.0 years) response (Figure 2A). The neutralizing capacity of the antibodies also did not correlate with the age of the donors (Figure 2B).

### 3.3. AAV-Induced Proliferation of Innate Cells and IFN-Beta Production after in Vitro Stimulation

PBMCs of seven donors (H1, H2, H3, H4, H6, H7, and C1) were tested for cellular immune responses to the three different AAV vectors. Donors H4 and H7 had low antibody titers, and H1, H2, H3, and H6 had intermediate titers (C1 had not been tested for antibodies), indicating that at least six of seven donors had an antibody response and, therefore, an adaptive immune response to AAV2. Three of seven had neutralizing antibodies (H2, H3, and H4). For stimulation, cells were incubated with the respective vector and tested for expansion of distinct cell populations and the production of IFN-β and IL-1β after 24 h and 4 days, respectively. Live “innate cells”, such as monocytes and DCs, as well as lymphocytes (T, B, and NK/NKT cells), were gated according to FSC (forward scatter) and SSC (side scatter) (Figure 3A). Data are shown as the “expansion index”, i.e., the population size of the stimulated cultures was divided by the size of the respective population in the medium control. This was used to obtain comparable data, since the baseline of the population size of the tested donors was individually highly variable. The stimulation of PBMCs in culture by each of the three vectors was generally very weak, and only those cell populations showing any expansion in response to the stimulation are shown in Figure 3. Neither T helper (CD3+/CD4+) nor cytotoxic T cells (CD3+/CD8+), in addition to CD56+NK or CD19+ B cells, showed any reactivity (Figure 3B). We could only observe a slight increase in CD19+ B cells and NK cells expressing the activation marker CD69+ in response to LPS, but not to any of the AAV2 capsids (Figure 3E). Further stimulation until day 4 did not reveal more responses of cells of the adaptive immune system.

Neither CD11+/CD14- DC, CD11-/CD14+ monocytes, nor the CD11+/CD14+ monocyte-derived DC population expanded their population size after 24 h of stimulation in response to any of the tested AAV vectors (Figure 3C). Among the monocyte population, only the CD14+ monocytes reacted to LPS with an expansion (Figure 3C). However, when cytokine production of innate cells was considered, IFN-β was produced by CD11c+ DC and CD11c-/CD14+ monocytes in response to all three capsids (Figure 3D). A high inter-individual variation of the CD14+ cell response to the capsids and a slight, statistically non-significant (*p* > 0.05), tendency for an increased response to AAV2.GL was noted (Figure 3D). A subpopulation of CD11c+/CD14+ monocyte-derived DC coproduced both IL-1β and IFN-β when stimulated with LPS or the AAV capsids. Again, a tendency for a higher response to variant AAV2.GL was observed. IL-1β production was only detected in response to LPS, but not to the AAV capsids (Figure 3D). None of the vectors induced an adaptive immune response in vitro, but all vectors induced the typical anti-viral IFN-β response by DC/monocyte-type innate cells (Figure 3D). Of note, since these assays were performed in antibody-free Panexin serum replacement, stimulation of innate cells by antigen-antibody-FcR binding can be excluded. 

### 3.4. Secretion of Cytokines and Chemokines Triggered In Vitro by AAV2 Variants

Next, we analyzed a panel of 48 cytokine and chemokine analytes from pooled supernatants collected daily from cell cultures of donors H1, H2, H5, H6, and H7 and stimulated with the AAV vectors. To evaluate the respective levels of secretion, we calculated the quotient of the secretion of the AAV-vector-stimulated cultures and the baseline secretion in the control medium. A secretion index of 2 or more was considered relevant. We observed a variety of cytokine and chemokine responses to vaccine antigens, such as PPD and TT, which were used as positive controls for the T cell response. In Figure 4, we display only 7 of the 48 tested cytokines or chemokines that showed a secretion index of 2 or more after stimulation with AAV vectors, as well as the lacking TNF-α and IFN-γ responses (Figure 4B); however, no statistically significant differences in the cytokine/chemokine responses were observed between the three tested AAV2 capsids. Of the seven detected cytokines, IL-6 showed the highest secretion in response to AAV2.NN (secretion index of 69) and less to AAV2 and AAV2.GL (secretion indices of 6.6 and 8.8, respectively). IL-8/CXCL8 and MIP-1α/CCL3 were both preferentially secreted in response to AAV2.GL and AAV2.NN with a secretion index of >10 and less than 2 for AAV2 (Figure 4). Gro-1α/CXCL1 and IL-1β were found at the highest levels in cultures stimulated with AAV2.NN and, to a lesser extent, in those stimulated with AAV2.GL or AAV2. LIF was slightly upregulated in response to AAV2.GL and less toAAV2.NN and AAV2. In contrast, VEGF secretion was only observed after stimulation with AAV2 (elevated four-fold). None of these changes in secretion reached the level of statistical significance (*p* < 0.05) and can only be regarded as tendencies.

In general, AAV2, as well as the two AAV variants, stimulated only production of monokines, but not T cell cytokines, and both capsid variants showed a tendency to better induce some cytokines and chemokines compared to AAV2, with AAV2.NN better stimulating the induction of Gro-α/CXCL1, IL-1β, and IL-6. Both variants better induced MIP-1α/CCL3 than AAV2, but none of the responses were significantly enhanced (Figure 4). 

## 4. Discussion

Although the eye is an immune-privileged organ, ocular gene therapy is often hampered by inflammation in the patient’s eyes, especially after intravitreal application of AAV vectors [26]. Due to the presence of less vector spreading and the immune privilege of the subretinal space, subretinal injection of AAV vectors appears to be less immunogenic, but carries the risk of adverse effects because retinal detachment is required to enable transduction of retinal pigment epithelium or the photoreceptors [26]. In order to overcome this limitation, next-generation AAV vectors with an enhanced tropism that transduce target cells in the outer retina (e.g., rod or cone photoreceptors) after intravitreal injection are currently being developed [15]. 

Vectors AAV2.GL and AAV2.NN harbor a 12-amino-acid insertion (different 7 mer targeting peptides flanked by alanine linkers) at insertion position I-587 of the AAV2 capsid. Insertions at I-587 separate R585 and R588, two important residues of the heparan sulfate proteoglycan binding motif, which is necessary for cell attachment [23]. 

We investigated how preexisting anti-AAV antibodies recognize and neutralize the novel AAV variants. Unfortunately, we had no access to sera of patients who were previously treated with ocular AAV gene therapy. Thus, any of the anti-AAV antibodies in our tested sera must derive from previous in vivo contacts of the blood donors with naturally occurring AAVs during adenoviral infection. Within the general population, the seroprevalence for AAV2 is variable, but generally quite high (up to >90%) [27]. AAV2-neutralizing antibodies are expected to bind surface-exposed positions, as indicated in Figure 5A and B, showing epitopes for the monoclonal antibodies A20 and C37-B, as well as epitopes identified by high-throughput screens of error-prone PCR-modified capsid libraries in the presence of neutralizing sera [28,29,30,31,32]. Neutralization is exerted by various mechanisms, such as direct competition for the receptor binding site on the viral capsid, steric inhibition by binding to a proximal epitope, and inhibition of endosomal escape or uncoating [33]. In agreement with Huttner et al., peptide insertion at I-587 had no major effect on general antibody binding in our study that compared AAV2 with AAV2.GL and AAV2.NN [34]. In contrast, we observed that neutralization of the engineered capsids was reduced compared to AAV2, albeit not in all tested sera. Since AAV2.GL and AAV2.NN differed from AAV2 by the peptide insertion in I-587, the lower sensitivity of affected sera is best explained by the presence of neutralizing antibodies that recognize epitopes at the capsid protrusions. As depicted in Figure 5C and D, our peptide insertion could mask the accessibility of the corresponding antibodies, thereby preventing their neutralizing effect. Comparing Figure 5C and D, the peptide insertion in AAV2.GL appears to cover the epitopes more efficiently, which might explain a more pronounced immune escape phenotype for AAV2.GL compared to AAV2.NN. 

Of note, not all tested sera contained neutralizing antibodies, and there was also no correlation between the total antibody responses and neutralizing antibodies, as previously shown [34]. Endemic infection with AAV affects between 50 and 96% of the population based on antibody positivity [37], which is also reflected by our cohort. Humoral immune responses to naturally occurring AAVs develop primarily in early life [38], which is line with the lack of any correlation between antibody titers, neutralizing antibodies, and age of donors. For ocular gene therapy, vectors are injected into the immune-privileged eyes and are, therefore, not primarily visible for the systemic immune system. Moreover, in the healthy eye serum antibodies are excluded from the inner compartments (e.g., retina) by the blood–eye barriers. Therefore, neutralizing antibodies might appear to be a lesser problem for ocular than for intravenously applied gene therapy [26]. However, under pathologic conditions leading to permeability or even breakdown of the blood–ocular barrier, the situation could be different. 

In addition to antibody binding, we investigated cellular immune responses to the novel AAV capsids in vitro. Again, we had no access to PBMCs of patients who were previously treated with ocular AAV gene therapy. Therefore, we only tested fresh peripheral lymphocytes of donors with variable levels of total and neutralizing anti-AAV2 antibodies. Treatment of donor PBMC cultures with the AAV vectors did not result in considerable proliferation of any tested subpopulation of the adaptive immune response. We speculate that either the uptake and processing of vector particles by the APCs in culture was insufficient or that none of our donors had an appropriate pre-existing cellular immune response. 

In contrast to antibody titers, the T cell memory responses rather quickly decline [39,40], which might explain why we could detect anti-AAV2 antibodies but no cellular responses to the capsids. 

High antibody titers and cellular responses to AAV2 do not correlate, as previously described by others who investigated in a similar experimental setting, and T cell responses to AAV are usually very low [37,41]. In general, AAVs are not strong inducers of immune responses [42]. 

Consistently with this, we observed a mild increase in innate immune cells, such as monocytes/macrophages (CD14+) and DCs (CD11+) producing IFN-β, but not IL-1β, in response to all three vectors. The CD11c+/CD14+ monocyte-derived DCs (moDCs) responded to LPS with IL-1β, but not IFN-β, while an increased subpopulation of moDC was detected that coproduced both cytokines in response to all three AAV vectors and to LPS. MoDCs are important players in viral infections and are very efficient in priming of cytotoxic T cells, cross-presentation, and secretion of cytokines [43,44,45]. Kuranda et al. described that AAV8 activated moDCs in human PBMC cultures and induced IL-6 and IL-1β [46], while we found an expansion of moDCs coproducing IL-1β as well as IFN-β, and we detected IL-6 secretion in stimulated PBMC cultures. 

Innate immune cells, such as monocytes or DCs, recognize antigens via their pathogen recognition receptor (PRR) and do not require processing and presentation. They can recognize whole viral capsids via the activation of TLR2 on their cell surface [47]. TLR2 recognizes repetitive protein subunits, which are typical for viral capsids [48]. As a consequence, AAV capsids might be sensed by TLR2 expressed by microglia, Müller cells, and RPE cells in the eye [49], regardless of AAV capsid modifications. 

Upon recognition, AAV vectors might induce uveitis, even in previously healthy eyes, as Tummala et al. showed in a mouse model [50]. The high incidence of uveitis (GTAU—gene-therapy-associated uveitis) occurring in up to 90% of patients’ eyes after ocular gene therapy and independent of pre-existing antiviral antibodies was also summarized and discussed by Chan et al. [26]. However, since we observed that TLR2-mediated sensing of AAV vectors depends on the vector dose, its reduction by increasing the target cell selectivity could help to diminish innate immune recognition in ocular gene therapy [51]. 

The detection of cytokines secreted in cultures incubated with the three tested AAV vectors revealed a different picture than the assay just focusing on proliferation. Analysis of the inflammatory cytokine release from the stimulated PBMCs showed no statistically significant differences between the three tested vectors, but a tendency of a better innate activation by the variant AAV2.NN or both novel variants in comparison to the parental AAV2, i.e., the response of monocytes, DCs and their production of IFN-β or of inflammatory cytokines and chemokines, such as CXCL1, CXCL3, IL-1β, and IL-6. 

Gro-α/CXCL1, MIP-1α/CCL3, and IL-8/CXCL8 are found to be elevated in viral infections and play contradictory roles, ranging from enhanced viral clearance to enhanced pathology, and they are attractants for macrophages, monocytes, and neutrophils; thus, they can trigger inflammatory reactions and blood–organ barrier breakdown, as shown for CXCL1 expression and the blood–brain barrier in viral encephalitis [52,53]. 

In conclusion, there was no evidence for an enhanced immune response to the peptide insertions at I-587. The preexisting immunity to AAV2 WT was generally very low in our tested donors, especially with respect to cellular immune responses. In various animal models for testing the transduction capacity of the vector variants, no obvious signs of ocular inflammation had been detected up to 3 weeks after injection of the vectors [23]. The significance of the decreased sensitivity of the engineered capsids to neutralizing antibodies needs to be investigated in patients in vivo. The increased tropism of the altered capsids for their retinal target cells could allow for dose reduction of the therapeutic vectors for ocular gene therapy and a reduced antigen load leading to decreased immune recognition or activation.

## Figures and Tables

**Figure 1 cells-11-01881-f001:**
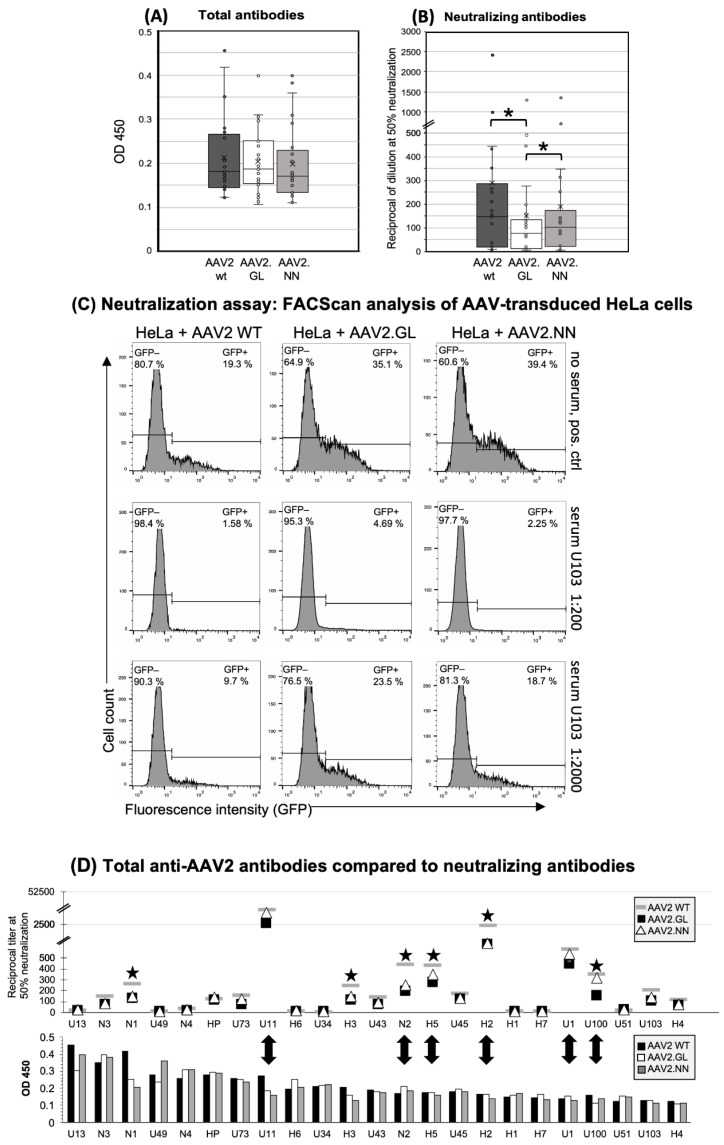
Human antibody responses to AAV2 WT, AAV2.GL, and AAV2.NN capsids. (**A**) Total Ig antibody responses (IgM, IgG, IgA) to the AAV capsids calculated from the ODs of all samples at a serum dilution of 1:400. *n* = 23. (**B**) Neutralizing antibodies shown as the reciprocal of the serum/plasma dilution at 50% inhibition of HeLa cell transduction (* *p* < 0.01). The small circles show the individual data points, the central line shows the median, the x represents the mean, and the whiskers above and below display the minimum and maximum within 1.5 interquartile range (IQR) of the lower and upper quartile. *n* = 23. (**C**) Representative histograms of the FACS analysis of AAV-transduced HeLa cells without serum (negative control, no neutralization, upper panel) and with serum U103 diluted 1:200 (strong neutralization, middle panel) and 1:2000 (reduced neutralization, lower panel). The left lines in each histogram show the gate of the non-transduced, eGFP-negative cells (percentage of cells in the upper left corner), and the right lines show the gate of the GFP-positive, vector-transduced cells (percentage of eGFP+ cells shown in the upper right). (**D**) Comparison of total and neutralizing antibodies for each donor. Upper panel: neutralizing antibodies, lower panel: total antibodies, shown as mean OD 450 of triplicates at sample dilutions of 1:200. Black stars mark samples with decreased neutralization of engineered capsids; double arrows high neutralizers and the respective total anti-capsid antibodies. *n* = 23 samples. The data shown are representative of repeated experiments. Statistics were performed using the Friedman test and Dunn´s Multiple Comparison test, and significance was defined as a *p* value of <0.05 (* *p* < 0.01).

**Figure 2 cells-11-01881-f002:**
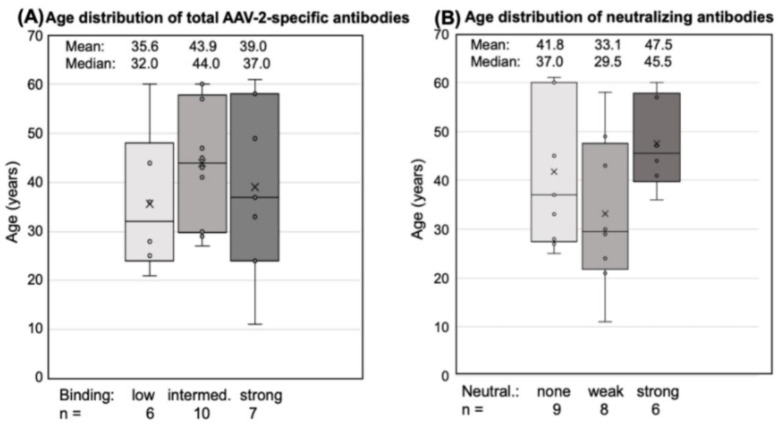
Age distribution of antibody responders. Total antibodies and neutralizing antibodies are shown for each donor. The small circles represent the individual data points, the central line shows the median, the x represents the mean, and the whiskers above and below display the minimum and maximum within 1.5 interquartile range (IQR) of the lower and upper quartile. (**A**) Total AAV2-specific antibodies: “low” binding was defined as mean OD 450 < 0.1 at 1:400 dilution, “intermediate” binding as a mean 0.1 < OD 450 < 0.24, and “high” binding as >0.24. (**B**) Neutralizing antibodies: “weak” neutralization needed 1:50 to <1:300 dilution at 50% neutralization, “strong neutralization” was defined as a serum dilution >1:300. Statistics were performed using the Friedman test and Dunn´s Multiple Comparison test; significant differences were not observed.

**Figure 3 cells-11-01881-f003:**
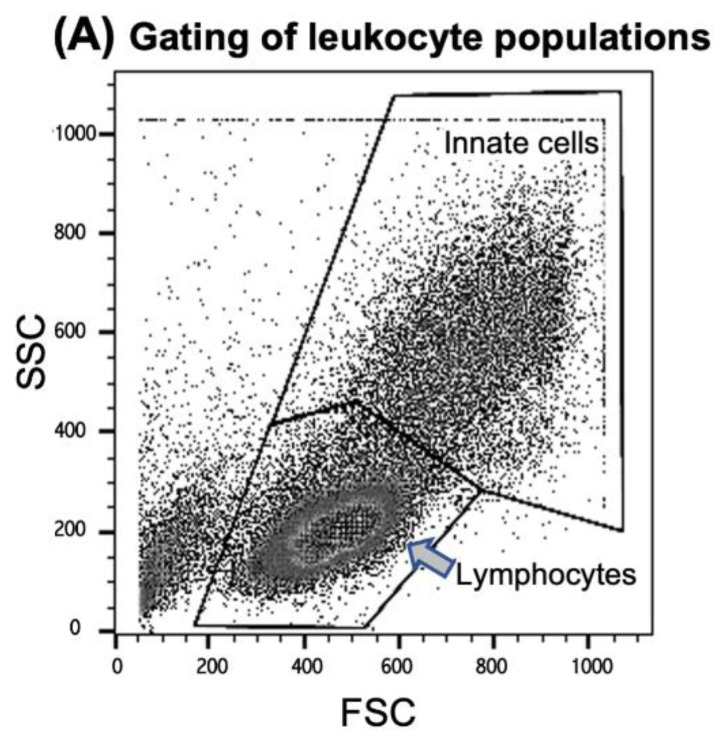
Expansion of cell populations after in vitro stimulation with AAV vectors. (**A**) Cells were gated for FACScan analysis according to FSC (forward scatter) and SSC (side scatter) in the “innate cells” population including monocytes/macrophages, DCs and granulocytes, and “lymphocytes”, including B, T, and NK/NKT cells (see the Materials and Methods section for more details). (**B**) CD3+ T cells (CD4+ Th, CD8+ cytotoxic T and CD56+ NK-T cells, CD3-CD56+ NK cells, and CD19+ T cells were analyzed from the lymphocyte gate. (**C**) CD11c+/CD14- DCs, CD11c+/CD14+ monocyte-derived DCs, and CD11c-/CD14+ monocytes were analyzed from the “innate cells” gate. (**D**) Surface staining for DCs and monocytes was combined with cytoplasmic staining for IL-1β and IFN-beta, respectively. The y-axis is cut at an expansion index of 12, as well as the expansion index of 40 from the LPS-simulated cells, and the CD11c-/CD14-/IL1β+ population is indicated in the respective column. (**E**) CD69-expressing lymphocyte subpopulations. The shown data are the means ± SD of the “expansion index” calculated as fold of the population size calculated from the population in the medium control. The results are shown as the mean expansion index of all tested donors (*n* = 6) after 24 h of in vitro stimulation, which was determined as the optimal time point from different stimulation experiments. The dashed lines mark twofold expansion of cell populations. Statistics were performed using the Friedman test and Dunn´s Multiple Comparison test; no significant differences were observed.

**Figure 4 cells-11-01881-f004:**
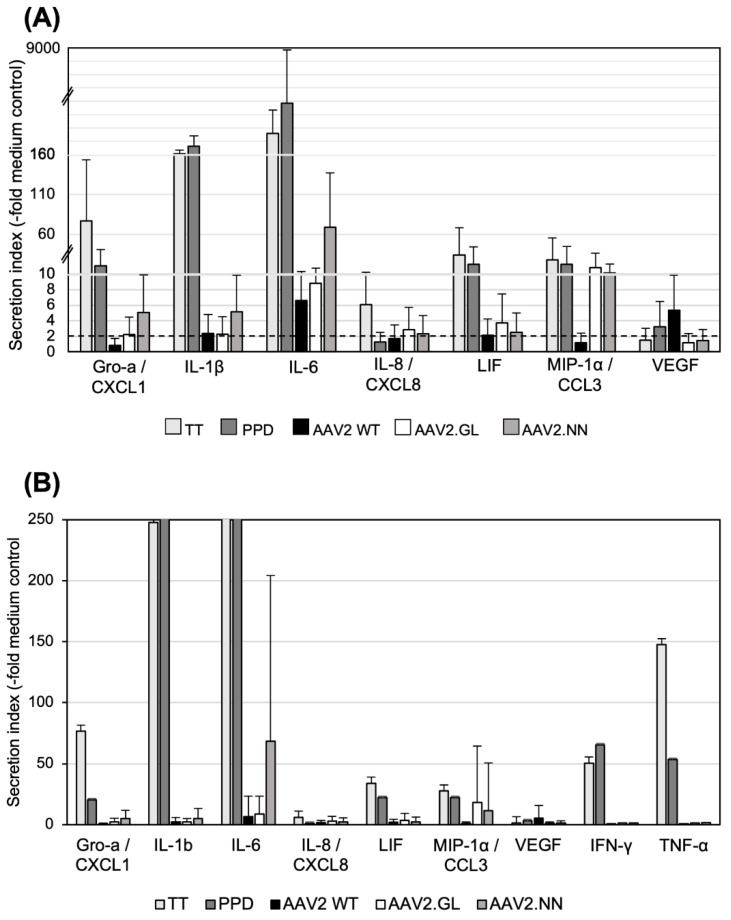
Cytokine secretion after in vitro stimulation with AAV vectors. (**A**) Innate cytokines/chemokines secreted by PBMCs stimulated in vitro. Pooled supernatants from three daily collections of stimulated PBMCs (*n* = 5; H1, H2, H5, H6, H7) were tested. Data are shown as mean “secretion indices” ± SD calculated as fold stimulation in cultures with TT, PPD, or AAV2 capsids as indicated and calculated from the secretion in cultures with medium only. The solid line marks the secretion index of 2. Note the different scales of the y-axis; transitions are marked by the dashed lines. (**B**) Same as (**A**), but with higher magnification of the y-axis to show the high variation in the cytokine responses among the different donors. No significant differences were observed between the AAV capsids. Statistics were performed using the Friedman test and Dunn´s Multiple Comparison test; no significant differences were observed.

**Figure 5 cells-11-01881-f005:**
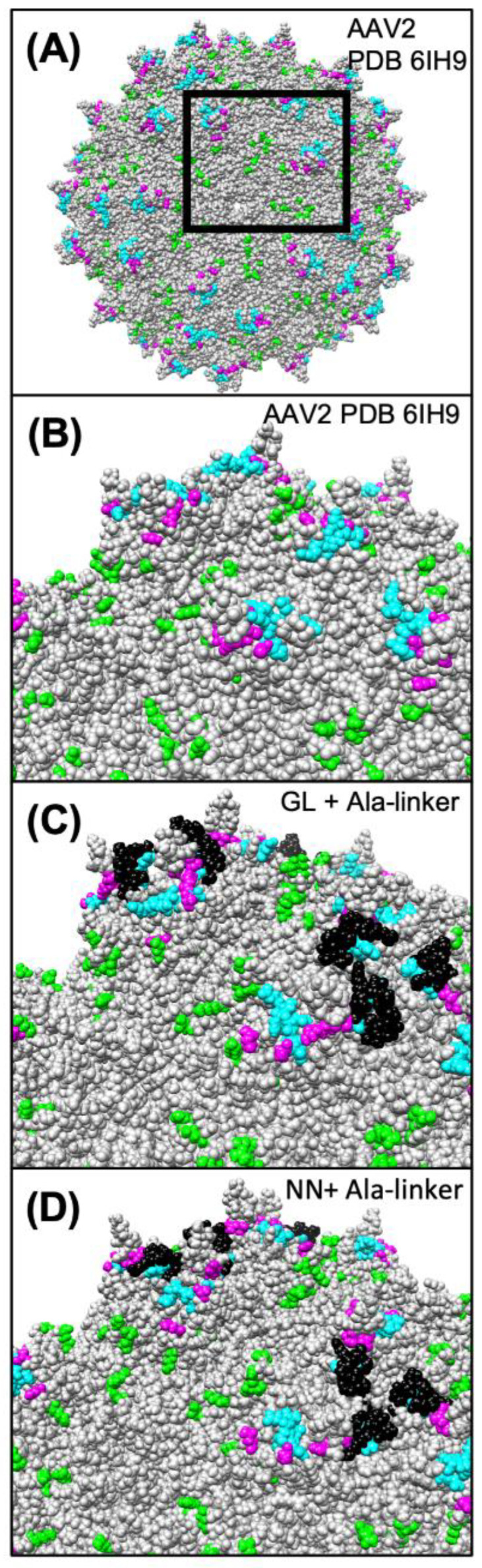
Modeling of the capsid epitopes of AAV2 WT and the variants AAV2.GL and AAV2.NN. Epitopes of known neutralizing monoclonal antibodies or serum (human and rabbit) of AAV2 are highlighted in different colors on the modeled structure of AAV2 WT (**A**,**B**) and the engineered capsids AAV2.GL (**C**) and AAV2-NN (**D**). Residues known to participate in the binding of the monoclonal antibodies A20 [28,29] and C37-B [28,30] are highlighted in green and cyan, respectively. Residues highlighted in magenta represent known epitopes for human and rabbit neutralizing antibodies [31,32,35]. Peptide insertions in AAV2.GL and AAV2.NN are highlighted in black. As is evident from the respective models, some of the epitopes are covered and occluded by the peptide insertions in AAV2.GL and AAV2.NN. Models were generated using the RoseTTAfold deep learning algorithm [36] available at https://robetta.bakerlab.org/ (accessed on 7 December 2021). The generated 3D models were visualized using the UCSF Chimera software (https://www.cgl.ucsf.edu/chimera/ (accessed on 7 December 2021).

**Table 1 cells-11-01881-t001:** Demographic data of blood donors.

No.	Donor	Gender	Age	P/S
1.	H1	female	60	P/S
2.	H2	female	41	P/S
3.	H3	male	43	P/S
4.	H4	female	28	P/S
5.	H5	female	60	P/S
6.	H6	male	45	P/S
7.	H7	male	60	P/
8.	HP	male pool	24	S
9.	N1	female	58	S
10.	N2	male	57	S
11.	N3	female	49	S
12.	N4	female	61	S
13.	U1	male	36	S
14.	U11	male	47	S
15.	U13	male	37	S
16.	U34	male	27	S
17.	U43	female	30	S
18.	U45	female	29	S
19.	U49	male	33	S
20.	U51	male	25	S
21.	U73	female	11	S
22.	U100	male	44	S
23.	U103	female	21	S
24.	C1	male	31	P

P: peripheral blood mononuclear cells, S: serum.

## Data Availability

Data can be obtained from the authors.

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
