# Peer review of "Immunogenicity of Novel AAV Capsids for Retinal Gene Therapy"

_cells, 2022, doi:10.3390/cells11121881_

Round 1
Reviewer 1 Report
The manuscript of Gehrke and collaborators, addresses a major issue regarding the assessment of immunogenicity of newly developed AAV vector capsids in view of ocular gene therapy applications.
AAV vectors have gained undisputable popularity as gene therapy vehicles for human applications, as demonstrated by the growing number of clinical trials and recent pharmaceutical developments. However, the need of high titer administration to reach therapeutic significant effects has unveiled a previously under evaluated intrinsic immunogenic and pro inflammatory risk of AAV vectors and put into question their safety record.
In this context, the analysis presented in the submitted manuscript represents an appropriate preliminary profile of immunogenic potential of candidate vectors for human therapy. These findings, validated by accurate and well-presented experimental evidences, add to a previously published description of the engineered capsid variants AAV2.GL and AAV2.NN, which showed an efficient widespread retinal transduction after intravitreal administration and a less immunogenic properties compared to AAV2 WT.
The present results confirmed that the two variants are less visible to pre-existing neutralising antibodies while they significantly elicit cellular immune response and cytokine production compared to AAV2.WT.
All these data are useful for the characterization of new capsid variants that promote a safer ocular gene therapy.
The limit of this work is that all the presented findings were obtained in vitro and only future in vivo pre clinical or possibly clinical administrations will tell if they represent meaningful prognostic evidence.
Although the significance of animal preclinical testing as a reliable prediction of the clinical immunogenic risk of AAV vectors is still debated, the present analysis could have been more exhaustive if would have included the characterization of immune/inflammatory response of intraocular injected animals.
Minor comment:
Consider revising the text of Neutralisation Assay method to make the procedure more understandable.
Author Response
We appreciate this positive evaluation on our manuscript and are thankful for the comments. We have carefully reviewed the manuscript and improved the description of the neutralization assay as suggested. We hope that the revised manuscript is now suitable for publication.
We thank the reviewer for the positive evaluation and helpful comments.
We agree that the description of the neutralization assay could be improved and have now changed the description of the neutralization assay methods to make it clearer.
Specifically, on page 4, line 152ff, we have introduced the explanation of how multiplicity of infection (MOI) was defined:
The MOI (multiplicity of infection, defining the ratio of vector particles to their target cells, determining the number of vector particles to obtain similar target cell transduction for different vector preparations) was determined for each AAV2 vector prior to the neutralization assay.
We also added a description of the normalization used to compensate for the differences in vector genome content between the preparations in line 159:
To compensate for differences in neutralization resulting from differences in genome content of the vectors, we adjusted as follows: Reciprocal serum dilution/(percentage of capsids with genome x 100).
In addition, we agree with the reviewer that our study was limited to in vitro studies. As noted by the reviewer, preclinical animal models have proven to be limited as a test system for evaluating anti-AAV immune responses when it comes to predict the outcome in humans. Nevertheless, our previously reported in vivo studies with our two novel AAV capsids (AAV2.GL and AAV2.NN) revealed improved transduction of the retina but no evidence of inflammation after intravitreal administration (Pavlou et al. EMBO Mol Med 2022). In the present study, we focused on the immune responses elicited in human blood cells and on the interaction with preexisting human neutralizing antibodies and therefore did not include further animal studies. We agree with the reviewer that future in vivo clinical administration is best suited to show whether our conclusion regarding a possible lower risk of immune stimulation by our capsid variants is correct (lines 504ff).
Reviewer 2 Report
This paper evaluates immunogenicity of two new vectors, AAV2.GL and AAV2.NN and compares them to wild-type AAV2. The immune response is a huge barrier to gene therapy as it necessitates the need for higher vector doses to be administered, which makes it more likely that there will be an inflammatory response. Modified AAV vectors may provided an option for those with pre-existing antibodies to AAV2, which is a large percentage of the population. Modifications of the AAV capsid to insert peptides to provide enhanced targeting to specific cell types, which is the case with these two novel AAV vectors also may allow for reduced vector titers. These new vectors have enhanced retinal tropism which may allow for more efficient transduction. The immune response to AAV vector administration provides a significant barrier to AAV delivery and studies of modified vectors are of interest to identify capsid mutants with enhanced tropism and better immune evasion. Samples were unavailable from individuals who have received ocular gene therapy, but this data would have been of interest as well. AAV ELISA was performed using novel AAV capsids and AAV2 to evaluate antibodies present in the individuals that participated in this study and whether or not they recognized the modified AAV capsids. Neutralization assays were performed to evaluate the neutralization capacity of human sera. In other assays, PBMCs were cultured in the presence of AAV and evaluated for cytokine expression. The data presented in this paper provides a better understanding of the antibody response to these novel AAV vectors with enhanced retinal tropism, as well as whether or not there was antibody neutralization. This paper would be of interest to the readers of this journal and the gene therapy audience as a whole.
Author Response
We appreciate this positive evaluation of our manuscript.